

# Genome-wide identification and characterization of *NAC* genes in *Brassica juncea* var. *tumida*

Longxing Jiang[1,*], Quan Sun[1,*], Yu Wang[1], Pingan Chang[1], Haohuan Kong[1], Changshu Luo[2] and Xiaohong He[1]

[1] Chongqing Key Laboratory on Big Data for Bio Intelligence, College of Bioinformation, Chongqing University of Posts and Telecommunications, Chongqing, China
[2] Chongqing Academy of Chinese Materia Medica, Chongqing, China
[*] These authors contributed equally to this work.

## ABSTRACT

**Background**. NAC (NAM, ATAF1/2, and CUC2) transcription factors play an important role in plant growth and development. However, in tumorous stem mustard (*Brassica juncea* var. *tumida*), one of the economically important crops cultivated in southwest China and some southeast Asian countries, reports on the identification of *NAC* family genes are lacking. In this study, we conducted a genome-wide investigation of the *NAC* family genes in *B. juncea* var. *tumida*, based on its recently published genome sequence data.

**Methods**. The *NAC* genes were identified in *B. juncea* var. *tumida* using the bioinformatics approach on the whole genome level. Additionally, the expression of *BjuNAC* genes was analyzed under high- and low-temperature stresses by quantitative real-time PCR (qRT-PCR).

**Results**. A total of 300 *BjuNAC* genes were identified, of which 278 were mapped to specific chromosomes. Phylogenetic analysis of *B. juncea* var. *tumida*, *Brassica rapa*, *Brassica nigra*, rice and *Arabidopsis thaliana* NAC proteins revealed that all *NAC* genes were divided into 18 subgroups. Furthermore, gene structure analysis showed that most of the *NAC* genes contained two or three exons. Conserved motif analysis revealed that *BjuNAC* genes contain a conserved NAM domain. Additionally, qRT-PCR data indicated that thirteen *BjuNAC* genes with a varying degree of up-regulation during high-temperature stress. Conversely, four *BjuNAC* genes (*BjuNAC006*, *BjuNAC083*, *BjuNAC170* and *BjuNAC223*) were up-regulated and two *BjuNAC* genes (*BjuNAC074* and *BjuNAC295*) down-regulated under low temperature, respectively. Together, the results of this study provide a strong foundation for future investigation of the biological function of *NAC* genes in *B. juncea* var. *tumida*.

## INTRODUCTION

Plant transcription factors are a group of regulators that inhibit the transcription rate of target genes by regulating their expression during plant growth and development to facilitate the response to various biotic and abiotic stresses (*Latchman, 1997*; *Jin et al.,*

Corresponding authors
Changshu Luo,
luochangshu7812@sina.com
Xiaohong He, hexh@cqupt.edu.cn

*2017*). Transcription factors form a transcriptional complex of regulatory genes by binding to DNA sequences or specifically interacting with other proteins (*Welner et al., 2015*; *Kim, Nam & Lim, 2016*). The NAC (NAM, ATAF1/2, and CUC2) transcription factor family is one of the largest families of transcription factors in plants. NAC transcription factors are named after no apical meristem (NAM) proteins found in *Petunia* hybrids (*Souer et al., 1996*), *Arabidopsis* transcription activation factor (ATAF1/2) proteins, and cup-shaped cotyledon (CUC2) proteins of *Arabidopsis thaliana* (*Aida et al., 1997*). The N-terminal of NAC proteins contains a conserved domain comprising 150–160 amino acid, which is further divided into five subdomains (A–E), whereas the C-terminal is a height diversity transcriptional regulatory region (*Welner et al., 2015*). Additionally, a group of NAC proteins, named NTL proteins (NAC with transmembrane motif 1-like), have been identified, which contain transmembrane regions with $\alpha$-helical transmembrane motifs (*Kim et al., 2006*; *Seo & Park, 2010*).

Genes encoding NAC transcription factors have been identified in many plant species, including *Arabidopsis*, rice (*Oryza sativa*), radish (*Raphanus sativus*), melon (*Cucumis melo*), *Brassica napus*, Tartary buckwheat (*Fagopyrum tataricum*), cacao (*Theobroma cacao*), and sesame (*Sesamum indicum*) (*Ooka et al., 2003*; *Nuruzzaman et al., 2010*; *Wang et al., 2015*; *Wei et al., 2016*; *Karanja et al., 2017*; *Zhang et al., 2018b*; *Liu et al., 2019*; *Shen et al., 2019*). In some plant species, the *NAC* transcription factors were known to play diverse roles. The first *NAC* genes identified in *Petunia* were shown to regulate the formation of embryos and flowers (*Souer et al., 1996*). Subsequent studies have shown that NAC transcription factors are involved in the response to many biotic and abiotic stresses (*Nakashima et al., 2012*; *Lv et al., 2016*), such as drought (*Wu, Wang & Wang, 2016*; *Hussain et al., 2017*), salt stress (*Wang et al., 2017*; *Alshareef et al., 2019*), floral morphogenesis (*Hendelman et al., 2013*), lateral root development (*He et al., 2005*), leaf senescence (*Guo & Gan, 2006*; *Kim, Nam & Lim, 2016*), embryo development (*Dalman et al., 2017*), hormone signaling (*Wang, Rashotte & Dane, 2014*), and xylogenesis, fiber development, and wood formation in vascular plants (*Ouyang et al., 2016*; *Sun et al., 2018*; *Zhang et al., 2018a*; *Yao et al., 2020*).

Currently, various NAC TFs are reportedly induced by temperature stress and participate in plant responses to cold and hot stress. Reports have shown that the expression of NAC TFs is significantly increased in tomato (*Liu et al., 2012*) and chickpeas (*Ha et al., 2014*) due to high and low temperatures. Overexpression of *GmNAC20* enhances salt and freezing tolerance in transgenic *Arabidopsis* plants. *GmNAC20* mediates stress tolerance by regulating the expression of COR (cold-responsive) genes, DREB1A/CBF3 (Dehydration Responsive Element-Binding Protein/C-repeat Binding Factor 3), and DREB1C/CBF2. *GmNAC20* can directly bind to the promoters of DREB1A/CBF3 and DREB1C/CBF2, thereby causing increased transcription of CBF3 and decreased transcription of CBF2. *MaNAC1* acts as a target gene of MaICE1 (Musa acuminata inducer of CBF expression 1), wherein *MaNAC1* protein interacts with MaCBF1 to regulate the cold tolerance of banana fruit (*Hao et al., 2011*). The overexpression of a *Malus baccata* NAC transcription factor gene *MbNAC25* also increases cold and salt tolerances of *Arabidopsis* (*Han et al., 2020*). NAC transcription factors play an essential role in high temperature stress. *Arabidopsis*

plants with *NAC019* overexpression are more heat-tolerant than the wild-type, and heat shock factors (HSFs) binding to the promoter element of heat shock proteins are the primary mechanisms for heat shock protein accumulation under heat stress. Heat stress can also regulate the expression of RCF2 (regulators of C-REPEAT BINDING FACTOR gene expression 2) (*Guan et al., 2014*), which interacts with and dephosphorylates *NAC019*; both proteins are necessary for heat induction of HSFs and thermotolerance. High temperature causes accumulation of $H_2O_2$ in plant cells, whereas positive feedback induces NTL4 gene expression in *Arabidopsis*. Reactive oxygen species (ROS) induce NTL4 gene transcription and NTL4 protein processing, wherein NTL4 gene and ROS constitute a positive feedback loop. High temperatures cause rapid accumulation of ROS in *Arabidopsis* and recycling of nutrients and metabolites in damaged tissues to meristems or newly formed leaves, thereby causing local programmed cell death and enhancing plants' survival rate (*Lee et al., 2014*).

Tumorous stem mustard (*Brassica juncea* var. *tumida*) is an economically important crop widely cultivated in China. *B. juncea* is an allotetraploid (AABB, $2n = 36$) that originated from hybridization between *Brassica rapa* (AA, $2n = 20$) and *Brassica nigra* (BB, $2n = 16$), followed by chromosome doubling. The enlarged stem of *B. juncea* var. *tumida* is used as an important seasoning, such as Fuling mustard. The growth of *B. juncea* var. *tumida* is affected by temperature, humidity, and light (*Liu, 1996*). The current research on *B. juncea* var. *tumida* is mainly focused on increasing yield via genetic breeding; however, few studies have been conducted on its internal molecular regulation mechanism of *NAC* genes. The genome data of *Brassica juncea* var.*tumida* has been published recently, which made it possible for us to find more information (*Yang et al., 2016*).

In this study, we used the HMMsearch software (*Potter et al., 2018*) to identify the NAC transcription factors in *B. juncea* var. *tumida* cultivar "Yongan xiaoye". The physiological and biochemical characteristics of *BjuNAC* transcription factors were analyzed, and a phylogenetic tree of *B. juncea* var. *tumida*, *Brassica rapa*, *Brassica nigra*, rice and *Arabidopsis thaliana* NAC proteins was constructed. Additionally, the expression of temperature-sensitive *BjuNAC* genes was analyzed at high and low temperatures, and the yield and quality of mustard tubers was investigated at high temperature (39 °C), low temperature (5 °C), and room temperature (25 °C; control). Overall, our results provide a theoretical basis of heat and cold stress tolerance in mustard.

## MATERIALS & METHODS

### Identification of NAC transcription factors

The genome sequence of *B. juncea* var. *tumida* was downloaded from the *Brassica* database (http://brassicadb.cn/#/Download/). The Hidden Markov Model (HMM) profile (*Eddy, 1998*) of the NAC protein (PF02365), which was downloaded from the Pfam database (http://pfam.xfam.org/) (*Finn et al., 2014*), was used as a query to identify the *NAC* genes in the *B. juncea* var. *tumida* genome using the local HMMER3.1 software, with *E*-value <1E-10 (*Potter et al., 2018*). Then, a species-specific HMM file of *B. juncea* var. *tumida* was generated using the HMMbuild protocol, and a new HMMsearch was performed. All putative *NAC* genes were identified using the NCBI CDD search (https://www.ncbi.nlm.nih.gov/cdd)

(*Marchler-Bauer & Bryant, 2004*); the conserved NAM domain was identified using SMART (http://smart.embl-heidelberg.de/).

## Multiple sequence alignment and phylogenetic analysis

The *AtNAC* protein sequences were downloaded from The *Arabidopsis* Information Resource database (https://www.arabidopsis.org/) (*Garcia-Hernandez et al., 2002*). Furthermore, the NAC protein sequences in rice were obtained from Rice Genome Annotation Project (http://rice.plantbiology.msu.edu/) (*Kawahara et al., 2013*). The NAC protein sequences of *Brassica nigra* were obtained using HMMsearch method above. And NAC proteins in *Brassica rapa* were downloaded from the Brassica database (http://brassicadb.cn/#/SearchTranscriptionFactorGene/?subfamily=NAC). All amino acid sequences of *BjuNAC, AtNAC, Brassica rapa, Brassica nigra* and rice proteins were aligned using Muscle alignment (*Edgar, 2004*), and a neighbor-joining phylogenetic tree was constructed using MEGA X, with 1,000 bootstrap replications (*Kumar et al., 2018*).

## Characterization of *BjuNAC* proteins and *BjuNAC* genes

The physicochemical properties (such as molecular weight and isoelectric point) of all putative *BjuNAC* proteins were determined using the ExPASy website (https://web.expasy.org) (*Artimo et al., 2012*). Conserved motifs were identified using MEME (http://meme-suite.org/) (*Bailey et al., 2009*), with the following parameters: maximum number of motifs = 15; search model = zero or one occurrence per sequence; motif length = 6–50 amino acids; and default settings for all other parameters. Information on the structure of all identified *NAC* genes was obtained from the GFF file, and the exon–intron map of *NAC* genes was constructed using the Gene Structure Display Server (GSDS) website (http://gsds.cbi.pku.edu.cn/) (*Hu et al., 2015*). Phylogenetic trees, motifs, and gene structure maps were drawn using the TBtools software (*Chen et al., 2020*).

## Chromosomal distribution cis-element and evolutionary analysis of *BjuNAC* genes

Information on the chromosomal location of *BjuNAC* genes was obtained from the GFF file using a Perl script (Supplemental 1) and then compiled manually. The chromosomal map of *BjuNAC* genes was drawn using the Mapchart software (*Voorrips, 2002*). MCSCANX (*Wang et al., 2012*) was used to analyze the collinearity and duplication events of *BjuNAC* genes. The synonymous substitution ratio (Ka/Ks) value of segmental and tandem duplication genes was used as the protocol sample Ka/Ks calculator in TBtools. PlantCARE (http://bioinformatics.psb.ugent.be/webtools/plantcare/html/) (*Lescot et al., 2002*) was used to perform cis-element analysis of the upstream regions (−1,500 bp) of *BjuNAC* genes.

## RNA extraction and qRT-PCR analysis

The expression pattern of *BjuNAC* genes in young leaves was studied under high and low temperatures. Seeds were sown in meteorite: soil (2:1) mix, incubated at 25 °C under 12-h light/12-h dark photoperiod in a growth chamber. After 5 weeks of seed culture, seedlings with the same growth conditions were placed at high temperature (39 °C) and

low temperature (5 °C) and cultivated for 0, 6, 12, 24, and 48 h; whole seedlings were used as the samples. After harvest, all samples were immediately frozen in liquid nitrogen and stored at −70 °C until needed for RNA isolation.

Total RNA was isolated from whole seedlings of *B. juncea* var. *tumida* using the Plant MiniBESTRNA Extraction Kit (TaKaRa Biotechnology, Dalian, China), and cDNA was synthesized using the PrimeScript™ RT Kit and gDNA Eraser (TaKaRa Biotechnology, Dalian, China). Then, quantitative real-time PCR (qRT-PCR) analysis was performed on the Bio-Rad iQ5 Real-Time PCR Detection System using the TBGreen® PremixEx Taq II Kit (Tli RNaseH Plus) RR820A (TaKaRa, Japan) and 14 pairs of gene-specific primers (Table S1), under the following thermocycling conditions: 95 °C for 30 s; 40 cycles at 95 °C for 5 s and 55 °C for 30 s; and 72 °C for 30 s. The *BjuActin* gene (*BjuB012485*) was used as an internal reference. Three biological replicates were performed for each gene, with each replicate containing three technical repeats. To analyze the expression change under different stress, the $2^{-\Delta\Delta Ct}$ method was applied (*Livak & Schmittgen, 2001*; *Rao et al., 2013*).

## RESULTS

### Genome-wide identification of *BjuNAC* genes

The HMM model file (PF02365) of NAC was downloaded from the Pfam database. A total of 336 *BjuNAC* genes were identified in the genome sequence data of *B. juncea* var. *tumida* using the HMMER 3.0 software. The redundant and non-NAM domain-containing genes were removed by domain identification using the SMART and NCBI websites, and 300 *BjuNAC* genes were extracted for further analysis. According to their chromosomal locations, 278 of 300 *BjuNAC* genes were named as *BjuNAC* 001–278, whereas the remaining 22 genes belonging to the scaffold or contig data were named as *BjuNAC* 279–300. Proteins encoded by *BjuNAC* genes varied in length from 164 amino acids (aa; *BjuNAC037*) to 1,432 aa (*BjuNAC236*) (Table S2), and their isoelectric point ranged from 4.19 (*BjuNAC012*) to 10.03 (*BjuNAC058*), with an average value of 6.50. The molecular weight (MW) of *BjuNAC* proteins ranged from 18.74 kDa (*BjuNAC037*) to 161 kDa (*BjuNAC236*), with an average molecular weight of 38.45 kDa. Among the NAC TFs studied, only 3 NAC proteins have MW >100 kDa and 36 NAC proteins are between 50 and 100 kDa. The MW of most NAC proteins is 20–50 kDa. The physiological and biochemical properties of all *BjuNAC* proteins are listed in Table S2.

### Phylogenetic analysis of *BjuNAC* proteins

We constructed a phylogenetic tree using the NJ method, with 1,000 bootstrap replications. On the basis of the classification of NAC proteins in other species (*Ooka et al., 2003*), all 300 *BjuNAC* proteins were divided into 18 subfamilies (Fig. S1). The results showed that *BjuNAC* proteins are as diverse as *AtNAC* and rice proteins, and 300 *BjuNAC*s were unevenly distributed in 18 subgroups. In our analysis, no *Brassica* NAC members from the subgroups ONAC001 and OsNAC3 were identified. The largest branch (NAC 2) contained 45 NAC proteins, whereas the smallest branch (TIP) contained only one NAC protein.

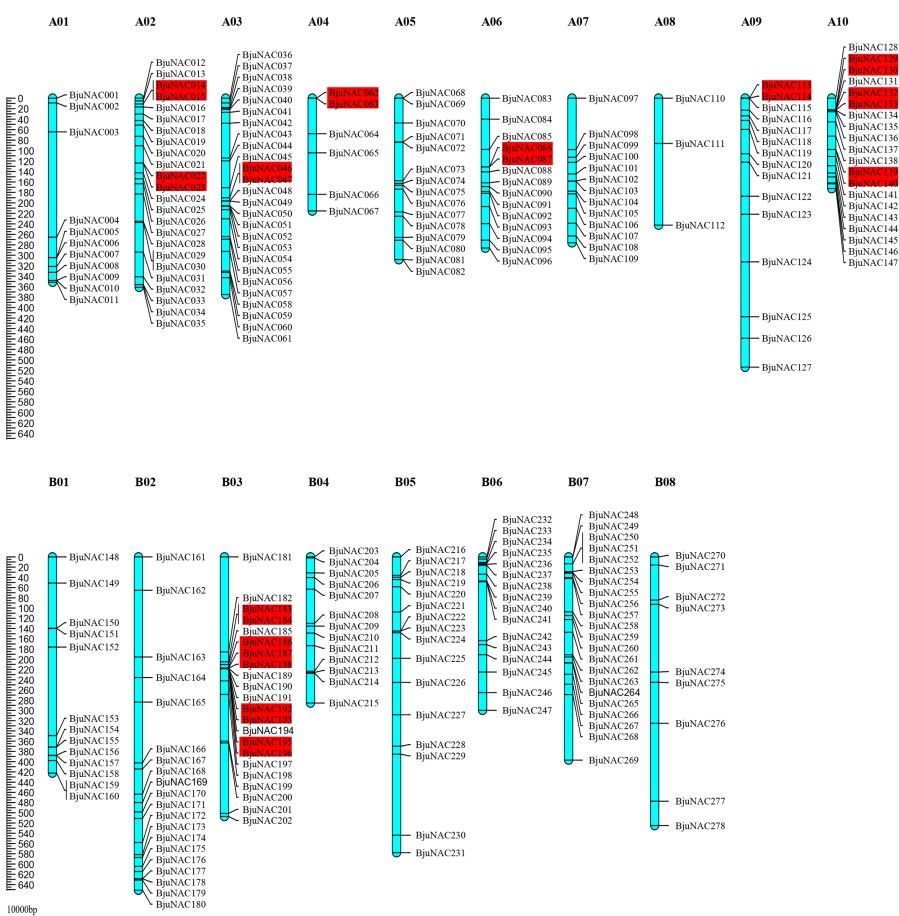

**Figure 1** **Chromosomal distribution of *BjuNAC* genes on the genetic map.** Graphical representation of locations for putative *BjuNAC* genes on each chromosome. The tandem duplicated genes were colored with red backgrounds. A01~A10, B01~B08 shows the chromosome numbers. Other 22 *BjuNAC* genes information belongs to scaffold or contig data were shown in Table S2.

Each subgroup contained different numbers of proteins, probably due to different genomic events during the evolution of *B. juncea* var. *tumida* and other species.

## Analysis of chromosomal distribution and duplication of *BjuNAC* genes

Out of 300 *BjuNAC* genes identified in this study, 278 were mapped to 18 chromosomes (A01–A10 and B01–B08). The highest number of *BjuNAC* genes (26) was mapped to chromosome A03; this was followed by 24, 22, 22, and 20 genes on chromosomes A02, B03, B07, and A10, respectively, and only 3 genes on chromosome A08 (Fig. 1). Distribution of *BjuNAC* genes on the chromosome is similar to *NAC* genes of *Brassica rapa* (*BrNAC*). It contains more *NAC* genes on chromosome A03 and A10, most of which are at the telomeric ends or near the centromere (*Liu et al., 2014*).

Collinearity analysis revealed that the mapped 220 *BjuNAC* genes formed 288 collinear pairs, including 14 tandem repeat pairs (Fig. S2). Additionally, 14 collinear pairs belonged to tandemly duplicated genes, including *BjuNAC14/15*, *BjuNAC22/23*,

*BjuNAC46/47*, *BjuNAC62/63*, *BjuNAC86/87*, *BjuNAC113/114*, *BjuNAC129/130*, *BjuNAC132/133*, *BjuNAC139/140*, *BjuNAC183/184*, *BjuNAC186/187/188* (three tandem repeats), *BjuNAC192/193*, *BjuNAC195/196*, and *BjuNAC280/282* (on the Contig11_2007833_2318034).

### Exon–intron gene structure and conserved motif analysis

To investigate the function of *BjuNAC* genes, we determined their exon–intron structure using the GSDS (Fig. S3). We also constructed a phylogenetic tree of all *BjuNAC* genes (Fig. S3A). The results showed that most of the *BjuNAC* genes (289 out of 300) contained introns, whereas the remaining 11 genes were intronless. Approximately half of the *BjuNAC* genes (159 of 300) contained two introns. *BjuNAC186* and *BjuNAC236* contained 18 introns, the highest number of introns among all *BjuNAC* genes (Fig. S3C).

To further analyze the structural diversity of *BjuNAC* proteins, conserved motifs were searched using the MEME program (Fig. S4). The distribution of motifs is shown in (Fig. S3B). Generally, *NAC* genes clustered within the same subgroup shared a similar motif composition (Fig. S5). Most of the NAC proteins included motif1 (representing subdomain A), motif4 (representing subdomain B), motif5 and motif3 (corresponding to subdomain C), motif2 and motif6 (corresponding to subdomain D), and motif7 (represented subdomain E). However, some of the conserved motifs were partially lost in proteins belonging to certain subgroups. For example, there were three *BjuNAC* proteins lost motif7 and three of them also lost motif 2, motif 3, motif 5 and motif6 in group VI. Half of the *BjuNAC* proteins in group IV; contained only motif1 and motif4. In group VII and VIII, almost all of *BjuNAC* proteins contained motif 8, especially in group VII, only BjuNAC276 lost motif 1 and motif 12 (Fig. S3B).

### Cis-acting element analysis

Cis-acting elements usually located within 1,500 bp sequence upstream of the transcription start site play a key role in regulating gene expression. Studies have shown that the expression of many genes depends on the presence or absence of specific cis-acting elements. To predict the function of *BjuNAC* genes, we analyzed the upstream regulatory sequence (−1,500 to +1 bp) of *BjuNAC* genes using PlantCARE. The results showed that the upstream sequences of most of the 300 *BjuNAC* genes contained cis-acting elements (Fig. S6) that respond to light, including the 3-AF1 binding site, AAAC-motif, ACA-motif, ACE, AT1-motif ATC-motif, ATCT-motif, Box 4, chs-CMA1a, GA-motif, GATA-motif, G-Box, and GT1-motif. Additionally, 77% of the *BjuNAC* genes contained anaerobic induction-responsive element and abscisic acid-responsive element (ABRE), 70% contained methyl jasmonate-responsive element (CGTCA-motif/TGACG-motif), 39% contained auxin-responsive element (TGA-box), 34% contained drought stress response element (MBS) and salicylic acid-responsive elements (TCA-element), 35% contained defense and stress response element (TC-rich repeats), and 36% contained low-temperature response element (LTR). The upstream sequences of only 22% and 11% of the *BjuNAC* genes contained circadian control regulatory elements (circadian) and seed development-specific elements (RY-element), respectively. A total of 110 genes have one

**Figure 2** **Expression levels of *BjuNACs* under cold and hot treatment by qRT PCR.** The number represented the treatment times (hours). The colour scales represent relative expression data. Note: *** $p < 0.001$; ** $p < 0.01$; * $p < 0.05$.

or more LTR cis-elements involved in low-temperature responsiveness. Four abiotic stress response elements, including ABRE, DRE, LTR, and TC-rich repeats, lay the foundation for further research on the potential regulatory mechanism of *NAC* genes in response to abiotic stress.

## Gene expression analysis under temperature stress

To further study the response of *BjuNAC* genes to abiotic stress, 14 *BjuNAC* genes enriched in the "temperature stress response" Gene Ontology term were selected for expression analysis under cold and heat stress treatments by qRT-PCR. Heat maps showing the expression of these 14 genes under the two stresses are shown in Fig. 2, the expression results are listed in Table S5. Most of the genes were up-regulated under the heat treatment, whereas some genes were induced by low-temperature stress. *BjuNAC006*, *BjuNAC083*, *BjuNAC170* and *BjuNAC223* were highly up-regulated under low-temperature conditions. Under the cold stress, most of the genes were up-regulated at 24 or 48 h, but some genes showed down-regulation after 48 h of cold treatment, for example, *BjuNAC295* was down-regulated under both cold and heat treatments.

## DISCUSSION

Tumorous stem mustard is an economically important food crop; however, its yield is affected by many abiotic stresses (*Cai et al., 2019*; *Li et al., 2019*). As one of the largest transcription factor families in plant species, the NAC family plays an important role in plant growth, development, and biotic and abiotic stress responses (*Seo & Park, 2010*; *Nakashima et al., 2012*). NAC transcription factors have been characterized at the genome level in many plant species (*Ling et al., 2017*; *Ahmad et al., 2018*; *Zhang et al., 2018b*; *Liu et al., 2019*). In this study, we conducted a comprehensive analysis of *BjuNAC* genes and the encoded transcription factors, including analysis of the gene structure, evolutionary relationships,

conserved motifs, chromosomal distribution, duplication events, and expression patterns under high and low temperatures.

A total of 300 *NAC* genes were identified in the *B. juncea* var. *tumida* genome, which is far greater than the number of *NAC* genes identified in other plant species. This maybe because the *B. juncea* var. *tumida* has a larger genome, which is an allotetraploid (AABB), resulting from the hybridization between diploid ancestors of *B. rapa* (AA) and *B. nigra* (BB), followed by spontaneous chromosome doubling (*He et al., 2020*). Therefore, the number of *NAC* genes in *B. juncea* var. *tumida* doubled. The number of NAC proteins in mustard tubers is more than that in *Brassica rapa* (204 members) (*Liu et al., 2014*) but less than that in NAC proteins in *Brassica napus* (410 members) (*Mohanta et al., 2020*), which indicates that the number of NAC proteins is positively correlated to the size of the *Brassica* genome. NAC family members were identified in *Brassica juncea* var. *tumida* (300 members) and *Brassica nigra* (212 members) (Table S3), using the HMMsearch method. Phylogenetic tree of NAC transcription factors in *Brassica juncea* var. *tumida*, *Brassica nigra*, *Brassica rapa*, *Arabidopsis* and rice, and all the NAC members were divided into 18 subfamilies. In the distribution of each subfamily, the number of *Brassica nigra* and *Brassica rapa* NAC members in subgroups was the same, but it was more in *Brassica juncea* than that of diploid. The copy number change of NAC family in *Brassica juncea* is not a simple doubling of the two diploids, but the loss or increase of genes between different subfamilies compared with the number of NAC in *Brassica nigra* and *Brassica rapa*. The isoelectric point of *BjuNAC* proteins varied from 4.19 to 10.03, with an average value of 6.50, which is consistent with most NAC TFs in *Brassica napus* that are acidic proteins. More than 68.3% of the *BjuNAC* proteins (205 out of 300) were acidic in nature, which may contribute to the acidic subcellular environment. Different NAC protein subgroups have different number of amino acids, protein MW, and isoelectric points. The versatility of NAC function may be the primary reason for the wide MW ranges of NAC proteins. Phylogenetic analysis revealed that all *BjuNAC* genes were divided into 18 subgroups. NAC proteins are involved in a variety of plant growth and developmental processes, such as seed germination, cell division, secondary cell wall biosynthesis, organ boundary and meristem formation, lateral root development, flowering, senescence, and iron balance regulation. Phylogenetic analysis can be used to predict gene function, which is very important for further analysis of gene function. NAM subgroups are involved in the formation of organ borders and meristem formation of plant, whereas ATAF1 subgroups are related to plant senescence and leaf curling. The overexpression of O*sNAC1* significantly enhanced drought resistance of transgenic rice in the field without phenotypic changes or yield reduction under severe drought stress conditions during the growth period (*Hu et al., 2006*). Although the expression level of *OsNAC7* in rice is extremely low or negligible in leaves, embryos, and callus, high expression levels are detected in stems and young panicles (*Kikuchi et al., 2000*). Expression of *OsNAC022* was induced by drought, high salinity, and abscisic acid (*Hong et al., 2016*). Salicylic acid and abscisic acid had no significant effect on *TaNAC08* gene expression (*Xia et al., 2010*). The TIP subfamily contains *BjuNAC074*, whose protein sequence is homologous to *Arabidopsis thaliana* TIP mRNA, wherein *AtTIP* gene is resistant to virus invasion in *Arabidopsis thaliana* (*Donze et al., 2014*). That implies NAC protein is

involved in the response to virus infection during plant vegetative development. *AtNAP* gene in the subfamily of NAC transcription factors is related to leaf senescence, and overexpression of the *AtNAP* gene causes premature senescence in *Arabidopsis thaliana*. According to the results of phylogenetic analysis, the *AtNAC3*, ATAF, and NAP subgroup proteins share a close relationship, and most of the published stress-related NAC family members are included in these three subgroups. *Arabidopsis* contains three proteins belonging to the *AtNAC3* subgroup (ANAC019, ANAC055, and ANAC072/RD26) and four proteins belonging to the ATAF subgroup (ATAF1/ANAC002, ATAF2/ANAC081, ANAC032, and ANAC102), all of which are NAC proteins. NAC proteins participate in the response to various abiotic and biotic stresses. In rice, the ANAC063 subgroup contained the highest number of NAC proteins, whereas the OsNAC8 subgroup was the smallest, with only three *BjuNAC* genes in *B. juncea* var. *tumida* is higher than that in other species, indicating that *BjuNAC* genes are more important for plant growth and development under various abiotic and biotic stresses.

Whole genome duplication has played an important role during plant genome evolution process, and segmental duplication is primarily responsible for the expansion of the *NAC* gene family. Collinearity analysis showed that 223 *BjuNAC* genes resulted from duplication, including 1 singleton, 27 dispersed, 9 proximal, 248 segmental, and 15 tandem duplications. These genes formed a total of 288 duplicate gene pairs, 50% of which originated from segmental duplication, indicating that segmental duplication represents the main mode of expansion of the *BjuNAC* gene family. At the same time, we found that 15 gene pairs were tandem repeats. Additionally, non-synonymous to synonymous substitution ratio (Ka/Ks) of segmental and tandem duplications was <1, indicating that the *BjuNAC* genes may have undergone purifying selection.

We also analyzed the exon–intron structure of all *BjuNAC* genes and conserved motifs in the encoded proteins. Molecular characterization of *BjuNAC* genes revealed the motif composition of the NAC transcription factor and is dispersed in a subgroup of gene structures that are not used. The *BjuNAC* genes clustered within the same subgroup generally shared a similar motif composition. The number of introns in *BjuNAC* genes varied from 0 to 18; only 11 *BjuNAC* genes were intronless, and approximately 50% of all *BjuNAC* genes contained two introns. These molecular features of *BjuNAC* genes are similar to the NAC genes of other plant species. Variation in the structure of *BjuNAC* genes indicates that the evolution of environmental stress response in these genes occurred through the acquisition or loss of introns.

We compared the NAC protein sequences of *Brassica rapa*, *Brassica nigra* and *Brassica juncea* respectively by the blast. Through the similarity of greater than 90%, we assumed that the two genes have homology. Finally, 151 *BniNACs* and 181 *BraNACs* were filtered. According to the analysis of chromosome distribution, genes deriving from chromosomes of *Brassica nigra* and *Brassica rapa* might have caused differences in the distribution of NACs in *Brassica juncea* var. *tumida* due to chromosomal rearrangements and structural variations within evolution. Among them, on chromosomes B1, B2 and B7 of *Brassica nigra*, there were 32, 20, 24 genes homologous to *BjuNAC* genes. On chromosome of *Brassica rapa*, 35 genes on chromosome A03 were homologous to *BjuNACs*, followed by

28 genes on chromosome A02, and the least only 2 genes (Table S4) on A08. *BniNAC*s chromosome distribution was shown in Fig. S7.

*B. juncea* var. *tumida* is sown from September to early October; however, late sowing significantly affects its effective tumorous stem yield. A temperature range of 15–20 °C is optimal for the germination of *B. juncea* var. *tumida* seeds. At the germination and seedling stages, changes in environmental conditions greatly influence the crop yield; sometimes only 3–5 days of difference on sowing date can cause 30% of the yield reduction of tumorous stem mustard (*Liu, 2014*). The region of Chongqing in southwest China experiences extreme weather in September and October, which has a greater impact on the germination of mustard seeds and the growth of seedlings. In Chongqing Fuling area, temperatures as high as 35 °C and 30 °C lasted for >5 days in September 2018 and 2019 and for 13 days in September 2019, respectively. Continuously high temperature decreases not only the seed germination rate but also the survival rate of seedlings.

The growth phase of mustard seedlings is from mid-October to late November. After 60 days of the growing season, mustard seedlings enter the stem expansion stage, which generally lasts from early December to mid-February (approximately 100 days) and is the most important growth phase of mustard plants. The optimal temperature for the stem expansion stage is 8 °C–15 °C, and the number of leaves is 25–45. At temperature <4 °C, the stem expansion of the mustard tuber is suppressed, which may lead to growth arrest. At −4 °C, frost damage is lethal for the mustard tuber (*Li, Dai & Zhan, 2015*). Long-term exposure to low temperature induces the plants to transition from vegetative growth to reproductive growth, resulting in bolting and consequently reduced yield. In 2008, the overall output of tumorous stem mustard was reduced by 30% because of excessive snow and ice (*Jian, 2019*).

The results of qRT-PCR showed that the expression levels of *BjuNAC006*, *BjuNAC073*, *BjuNAC083*, *BjuNAC223*, *BjuNAC241*, and *BjuNAC170* increased significantly over time, regardless of the heat or cold stress, indicating that these genes may be involved in response to temperature stimuli. *BjuNAC073* and *BjuNAC006* are homologous to *BnaNAC103* and *BnaNAC55*, respectively. *BnaNAC103* (*Niu et al., 2014*) and *BnaNAC55* (*Niu et al., 2016*) respond to multiple signals, including heat, cold, salicylic acid, and a fungal pathogen, *Sclerotinia sclerotiorum*. Simultaneously, they play a role in activating the promoter activity of five genes encoding ROS-scavenging enzymes or mediating defense responses. Expression of *BjuNAC083* is consistent with *NAC* gene expression (similar sequence with *Bra029201*) under heat and cold stress in *Brassica rapa* (*Liu et al., 2014*). *BjuNAC006* (sequence similar to *ANAC055*) and *BjuNAC083* (sequence similar to *ANAC019*) transcription factors bind specifically to 5′-CATGTG-3′ motif and respond to water shortage at high temperatures (*Tran et al., 2004*). The expression level of *BjuNAC178* reached a peak at 6 h, then declined under cold stress and gradually increased under heat stress, which indicated the role of *BjuNAC178* in temperature stress response. The expression level of *BjuNAC240* reached a peak at 24 h, but did not vary much under cold stress. BLAST (basic local alignment search tool) comparison showed that *BjuNAC240* is homologous to *BnaNAC60*, positively modulating programmed cell death and age-triggered leaf senescence (*Yan et al., 2020*). The expression of *BjuNAC112* and *BjuNAC184* was the highest at 24 h under heat stress,

but did not change significantly under cold stress, indicating that these genes may be involved in the high temperature stress response. The expression level of two *BjuNAC* genes (*BjuNAC074* and *BjuNAC295*) was down-regulated under low temperature, and they may respond to cold stress. Nonetheless, the specific biological functions of *BjuNAC* genes need to be verified experimentally.

## CONCLUSIONS

In this study, we identified 300 *NAC* genes in *B. juncea* var. *tumida* and characterized their gene structure, chromosomal distribution, phylogenetic relationship, conserved motifs, and expression pattern in response to abiotic stress. *BjuNAC* genes showed differential expression patterns under high- and low-temperature stresses. Overall, our findings provide a strong basis for further investigation of the biological functions of *BjuNAC* genes. This information could be used to increase the yield of *B. juncea* var. *tumida*, thus increasing its food value and economic importance.

### Funding

This work was supported by grants from the Science and Technology Research Program of Chongqing Municipal Education Commission (Grant No. KJQN201800609) and Natural Sciences Foundation of Chongqing (Grant No. cstc2015jcyjA0752). The funders had no role in study design, data collection and analysis, decision to publish, or preparation of the manuscript.

### Grant Disclosures

The following grant information was disclosed by the authors:
Science and Technology Research Program of Chongqing Municipal Education Commission: KJQN201800609.
Natural Sciences Foundation of Chongqing: cstc2015jcyjA0752.

### Competing Interests

The authors declare there are no competing interests.

### Author Contributions

- Longxing Jiang performed the experiments, analyzed the data, prepared figures and/or tables, and approved the final draft.
- Quan Sun conceived and designed the experiments, authored or reviewed drafts of the paper, and approved the final draft.
- Yu Wang performed the experiments, prepared figures and/or tables, authored or reviewed drafts of the paper, and approved the final draft.
- Pingan Chang conceived and designed the experiments, prepared figures and/or tables, authored or reviewed drafts of the paper, and approved the final draft.
- Haohuan Kong performed the experiments, prepared figures and/or tables, and approved the final draft.

- Changshu Luo analyzed the data, authored or reviewed drafts of the paper, and approved the final draft.
- Xiaohong He conceived and designed the experiments, prepared figures and/or tables, and approved the final draft.

## Data Availability

The raw measurements are available in the Supplementary Files.

## Supplemental Information

Supplemental information for this article can be found online at http://dx.doi.org/10.7717/peerj.11212#supplemental-information.

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
