# Peer review of "Genome-wide identification and characterization of NAC genes in Brassica juncea var. tumida"

_PeerJ, doi:10.7717/peerj.11212_

## Round 0.1 · original submission · Minor Revisions

Dear Dr. He,

We have received reviews of your manuscript, which you may view at the end of this letter. On the basis of the advice received and my own evaluation, I believe that, pending revision, this article may ultimately be acceptable for publication PeerJ.

The two knowledgeable reviewers, while favorably impressed by the work, made many excellent comments and suggestions for improvement. I agree that following these comments will great improve the manuscript. Please take care at revising your discussion and including the phylogenetic comparison.

Thank you very much for giving us an opportunity to review this work. I look forward to receiving the next version.

Sincerely,
Vincent Courdavault

·

Basic reporting

NAC proteins are a large family of transcription factors, participating in diverse physiological and developmental roles as well as a defence response against biotic and abiotic stresses. Authors report the genome-based survey of NAC gene family in Brassica juncea var. tumida.

In addition to their in silico study, they made expression analyses of selected NAC genes in different temperature stresses. This study may be the first step for further analysis of NAC’s function and application in the tumorous stem mustard

Experimental design

The authors followed a classic methodology for this kind of study. The experimental part (wet bench) could be more complete.

Validity of the findings

To enhance the impact and novelty of their study, the current version of the manuscript needs revision.

Additional comments

The work of Jiang et al. presents an in silico genome-wide analysis of the NAC proteins in the crop plant in the tumorous stem mustard. The authors have used published genome sequence information from in Brassica juncea var. tumida to mine for NAC genes where 300 have been found. They performed classification and phylogenetic analysis with other species. Overall the manuscript describes a very important and interesting development in this particular research field. Moreover, it gives some interesting scientific perspective on this topic, which is a major concern for the next years (ie, global warming). NAC gene family is assumed to have different roles in diverse plant function and process. However, authors do not discuss the precise and specific role of BjuNAC gene family. Authors should emphasize and clearly discuss the role and function of such genes, it is not clear how BjuNAC genes are regulated under different stress condition. Yet authors may think of following suggestions:

- Your introduction needs more detail. I suggest that you improve the description at lines 58- 67 to provide more information regarding NAC proteins and temperature stress.
- Line 69: move(AABB, 2n = 36) after allotetraploid
- Line 72-75: if possible, add some references.
- Line 76-90: this paragraph can be more concise or moved to the Discussion part
- Line 93: typo (a space is missing before the reference)
- Methods: a description of your script (supplemental?) can be useful (line 140)
- Line 161: why did you choose Actin as housekeeping gene?
- Results: it may be interesting to add HSE cis-element because they are key element in the heat-shock response (line 221)
- Discussion should be carefully revised. The manuscript needs to further discuss and enrich the functional role of BjuNAC genes, compare results with the existing literature (what is similar, what is different), explain why did you choose treatment of 24h, 48h. It is a very strong treatment affecting many physiological processes. Authors need to be careful with their interpretation. Performing qRT-PCR on the hallmark of HS/cold stress in theses conditions may add strength to the manuscript (in supplemental?)
- Size-wise, BjuNAC proteins varied a lot. Can you talk a bit about that? What would be the adaptive benefits of such diversity?
- Line 282: add “in rice, “ before “The OsNAC7 subgroup”
- 289-296: Can you elaborate more what are the function of these NAC during the heat and cold stress?
- Line 303-305: Can you detail in the Methods part of you obtained this result?
- Line 309. I will be more careful with the statement. Only a few BjuNAC genes were experimentally tested.

Figures:
- Fig1. Because you compare NAC proteins in different species, it can be useful to add other crop species such as rice (and wheat?).
- Fig4. Adding labels 39°C, 5°C can improve the figure.
- Adding all clustalW alignment in supplemental figures may be interesting as we can see if the sequences are similar or not.

English and typo error: a careful proofreading may correct the few mistakes in the manuscript.

Reviewer 2 ·

Basic reporting

It is a nicely written manuscript and literature review is upto date.

Experimental design

Adequate, but a comparison with basic diploid progentors would have been very useful.

Validity of the findings

Conclusions are adequate and based on data generated. I missed comparison with diploid progenitors.

Additional comments

It is a very informative and well compiled. However, one would have liked to include phylogenetic comparison of BjuNAC proteins with those from Bni NAC and Br NAC . Same should be true for chromosome distribution, copy number variation and gene expression analysis across ploidy.

---

## Round 0.2 · Minor Revisions

Dear Dr.He,

Based on your revised manuscript that addresses the main concerns raised by the two reviewers, I'm pleased to inform you that your article is almost ready to be for publication.

However, a Section Editor, has commented and said:

"1) The juncea sequencing paper (Yang et al, 2016) needs to be cited. +++ 2) Please carefully double check the manuscript for grammar mistakes. for example, line 34 "ruie" should be "role". +++ 3) Abstract overstates conclusion. It is incorrect to conclude, from expression data alone, that the NAC genes play a crucial role in temperature stress. +++ 4) Statistical analysis of the qRT-PCR results must be done to determine if the changes in expression are significant. One common method is here: https://pubmed.ncbi.nlm.nih.gov/25558171/, which should be followed by t-test, ANOVA, linear regression, or equivalent."

Please address these points and resubmit.

Thanks for submitting to PeerJ.

best,
vincent

---

## Round 0.3 · accepted · Accept

Dear Dr. He,

Following the receipt of your revised version, I'm pleased to inform you that your manuscript has been accepted for publication.
Thanks for submitting your work to PeerJ.

Best regards,
vincent